# Light controlled 3D micromotors powered by bacteria

Gaszton Vizsnyiczai[1], Giacomo Frangipane[1], Claudio Maggi[2], Filippo Saglimbeni[2], Silvio Bianchi[2] & Roberto Di Leonardo[1,2]

Self-propelled bacteria can be integrated into synthetic micromachines and act as biological propellers. So far, proposed designs suffer from low reproducibility, large noise levels or lack of tunability. Here we demonstrate that fast, reliable and tunable bio-hybrid micromotors can be obtained by the self-assembly of synthetic structures with genetically engineered biological propellers. The synthetic components consist of 3D interconnected structures having a rotating unit that can capture individual bacteria into an array of microchambers so that cells contribute maximally to the applied torque. Bacterial cells are smooth swimmers expressing a light-driven proton pump that allows to optically control their swimming speed. Using a spatial light modulator, we can address individual motors with tunable light intensities allowing the dynamic control of their rotational speeds. Applying a real-time feedback control loop, we can also command a set of micromotors to rotate in unison with a prescribed angular speed.

[1] Dipartimento di Fisica, Università di Roma 'Sapienza', Roma I-00185, Italy. [2] NANOTEC-CNR, Institute of Nanotechnology, Soft and Living Matter Laboratory, Roma I-00185, Italy. Correspondence and requests for materials should be addressed to R.D.L. (email: roberto.dileonardo@uniroma1.it).

Dense suspensions of swimming bacteria display striking motion that look extremely vivid compared to the thermal agitation of colloidal particles of comparable size. These suspensions belong to a wider class of non-equilibrium systems that are now collectively referred to as active fluids[1]. A consistent portion of research in active matter physics deals with the fundamental aspects underlying some distinctive properties of these systems, such as the emergence of collective behaviour and rectification[2,3]. From another intriguing perspective, however, active fluids can be looked at as a special kind of fuel: a small droplet of an active fluid can be used to propel micromachines inside miniaturized chips, with no need of external driving fields or control. In these fluids energy is directly present in a mechanical form and the challenge is that of designing microstructures that are able to rectify the noisy and disordered motion of active particles into a reproducible and smooth directed movement. The first studies in this direction focused on the development of bacteria-propelled microrobots[4–9] with potential applications for drug and cargo delivery. These bio-hybrid microrobots were based on microfabricated structures or microbeads with a biochemically functionalized surface. Swimming bacteria adhere to the surface and act as micropropellers, which can also be switched on and off by using chemical[4] or light signals[5]. Yet, the motion of these microrobots is random, so that directed motion can only be achieved by an external feedback action through optical stimuli[7] or magnetic fields[8]. Other studies focused on creating bacteria-propelled rotary micromotors. The first example was reported in ref. 10, where gliding bacteria moving in narrow tracks were biochemically bound to a rotor that was pushed at rotational speeds of about 2 r.p.m. Still, the motion was intermittent, with continuous rotating phases lasting typically for a minute, and a consistent fraction (16%) of rotors were spinning in the opposite direction. Few years later a quite different approach was born based on the idea that, due to broken detailed balance in active baths[3], unidirectional motion can also be achieved by spontaneous rectification effects induced by objects having asymmetric shapes[11–14]. These bacterial ratchets do not require surface functionalization or external fields, they only rely on their morphology to induce self organization of bacteria into partially ordered configurations that apply a net force or torque on the object. These rotating flat structures roam unconstrained over an interface and this limits their usability in practical applications. Furthermore, the instantaneous arrangement of driving bacteria is highly stochastic resulting in temporal speed fluctuations that are comparable to the mean. For the same reason, a wide distribution of angular speeds is observed among different rotors.

Here we use two-photon polymerization to build composite 3D structures that autonomously capture bacteria into precise configurations exerting a stable and large torque on a rotor constrained to revolve around a fixed axis. We show that such structures can be fabricated in large arrays and that they independently rotate with a high and smooth angular speed. In addition, using a smooth swimmer *E. coli* strain expressing a light-driven proton pump, we show that the speed of micromotors can be controlled through the intensity of illumination light. We can tune the speed of individual micromotors by independently adjusting their illumination levels with a spatial light modulator. Furthermore, through a real-time feedback control loop, we command a set of micromotors to rotate in unison with a prescribed angular speed.

## Results

**3D micromotor design.** Figure 1a shows the computer model of our micromotor design. The structure has three component parts. The rotating unit (appearing in green) has an external radius of 7.6 μm and a thickness of 3.7 μm. Its outer rim features 15 microchambers, each capable of accommodating one single cell body while leaving the entire flagellar bundle outside for maximal propulsion. The number of chambers a single rotor could carry is limited by the constraint that each chamber should be wide enough to fit one cell. Therefore the number of chambers is maximal when they are radially oriented but this configuration would also result in a zero torque. If we tilt the chambers by an angle θ the torque exerted by each cell will increase with sin θ while the number of chambers will decrease as cos θ. As a result the total torque would go as sin θ cos θ and have a maximum at θ = 45° which is the actual angle chosen in our design. To reduce fabrication time, the outer ring of the rotor is connected by four radial spokes to a central ring that is free to rotate around a vertical axis shown in blue. The top part of the axis has a wider cross-section to prevent rotor escape. A bottom platform on the axis keeps the rotor at a minimal height of ∼3 μm, thus reducing hydrodynamic friction with the solid substrate. Moreover, the elevated position keeps the rotor out of the high bacterial concentration layer that forms on the substrate surface[15–17]. There, random collisions with free swimming bacteria would disturb the rotor's operation making it slower and less stable. To facilitate the capture of bacteria by the rotor, we built a radial ramp structure (red component) that collects bacteria swimming on the bottom surface and directs their trajectories upwards to the rotor's microchambers. However, with a bare ramp only a fraction of bacteria would be sent along trajectories that reach the

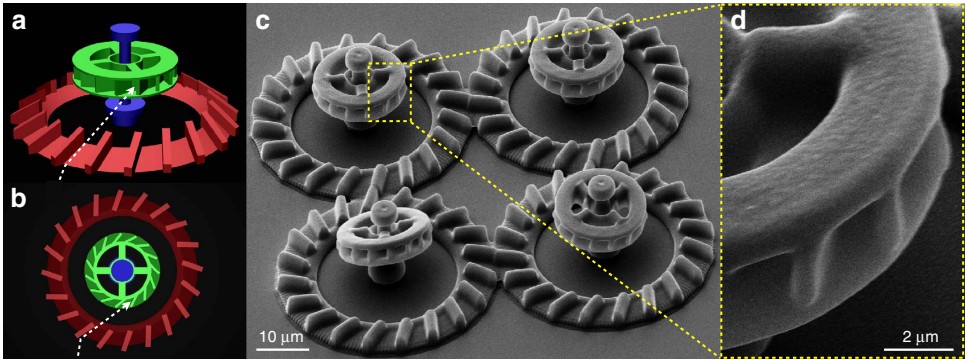

**Figure 1 | Design of 3D micromotors. (a,b)** 3D model of the micromotor structure. Colours highlight distinct component parts: ramp (red), axis (blue) and rotor (green). The dashed white line schematically depicts the trajectory of a cell guided by the ramp structure into a rotor microchamber. **(c,d)** Scanning electron microscope images of the 3D micromotors. **(c)** shows a bird's eye view on a set of four micromotors. **(d)** shows a close view of microchambers.

tilted microchambers with the correct orientation to enter it. To overcome this problem we placed a series of barriers on the ramp. Incoming bacteria will align and slide along the barrier leaving the ramp on a trajectory that is very likely to intercept the rotors edge with an angle matching the orientation of the microchambers (see dashed white line on Fig. 1a,b). All experiments were performed with non tumbling strains (smooth swimmers) which can be stably trapped in an an empty hole by a constant flagellar thrust. We used two-photon polymerization[18,19] to fabricate micromotors from SU-8 photoresist on top of a microscope coverglass. Details of the actual structures, as seen by scanning electron microscopy (SEM), are shown in Fig. 1c,d).

**Self-assembly of hybrid micromotors.** We first immerse the structures in clean motility buffer ($\sim$500 μl) and make sure that none of the rotors is stuck to the supporting axis. We then add $\sim$100 μl of a dense ($OD_{590} = 0.8$) cell supension resulting in typical surface densities of cells over the coverslip of about 0.015 cells per μm$^2$. Within few minutes after the addition of bacteria, the rotors start to capture cells and rotate. This self-assembly mechanism is very robust, practically every single rotor spins even in large and dense ensembles such as the array of 36 rotors shown in Fig. 2a and Supplementary Movie 3. By transforming a RFP-expressing plasmid in our strain we can use epifluorescence microscopy to visualize the cell bodies captured inside the microchambers (Fig. 2c,e and Supplementary Movies 1 and 2). To characterize the dynamics of this self-assembly process we recorded a sequence of bright-field clips (2 s) intercalated by short clips (0.1 s) in epifluorescence mode throughout the entire fill up process. Bright-field clips are used to track rotational speeds of individual rotors while simultaneously keeping track of the corresponding number of trapped cells observed in fluorescence images. Figure 3a shows the number of captured bacteria as a function of time. The average number of occupied microchambers (shown as red crosses) reaches the 90% of its maximal value (13.5) after only 2 min. The overall time behaviour can be very well fitted by an exponential law with a time constant of 49 s and a saturation value of 14.4 (96%). We find that the rotational speed increases linearly with the number of captured cells (Fig. 3b) with each cell contributing a speed increment of 1.2 r.p.m.

**Characterization of rotational dynamics.** Using a feature detection algorithm[20] we extract the orientation angle of rotors from image sequences acquired at 50 fps under bright-field illumination. Figure 4a shows the cumulative rotation angle for 16 rotors as a function of time. The average angular speeds of individual rotors are distributed in a narrow range around a mean value of 17.8 r.p.m. (1.4 r.p.m. s.d.). The grey line in Fig. 4b shows

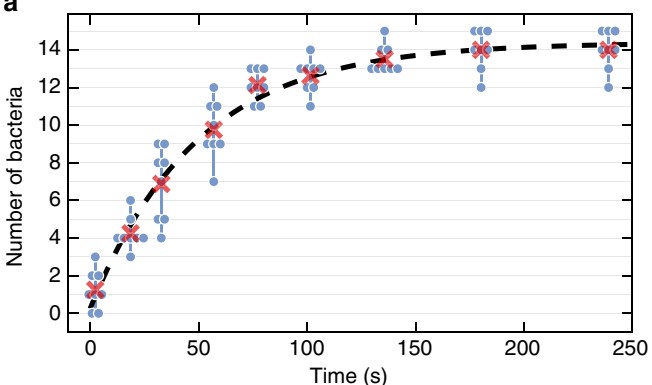

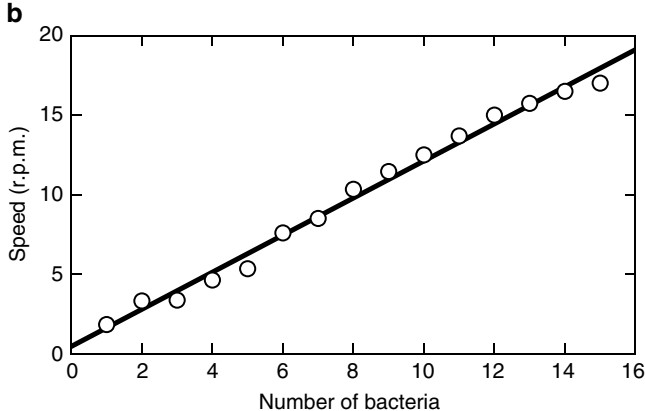

**Figure 3 | Self-assembly dynamics.** (**a**) The number of captured bacteria as a function of time is plotted as blue disks for 8 micromotors. Groups of blue dots connected by vertical lines refer to the same time instant. The average over the 8 micromotor group is plotted with red crosses and fitted to the exponential law shown as dashed line ($\tau = 49$ s). (**b**) Rotational speed as a function of the number of captured bacteria (open circles). Linear fit (black line) gives a slope of 1.2 r.p.m. per cell.

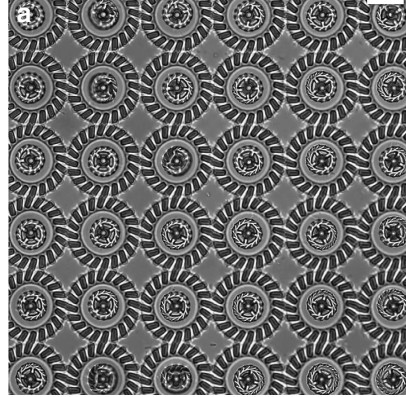
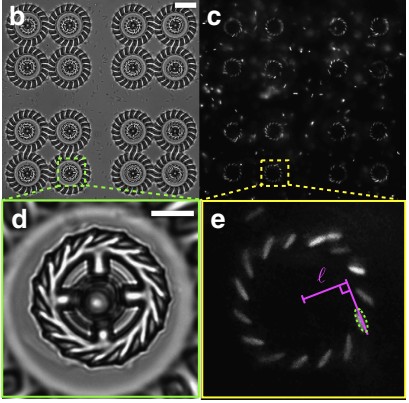

**Figure 2 | Micromotors in a bacterial suspension.** (**a**) Bright-field microscopy image of 36 rotating micromotors (Supplementary Movie 3). The scale bar is 20 μm. (**b,c**) Array of 16 rotors used to characterize the rotational dynamics (Fig. 4). Cell bodies are clearly visible in fluorescence (**c**) showing the high occupancy fraction of microchambers. The scale bar is 20 μm for both (**b,c**). (**d,e**) Zoomed view on one of the rotors in **b,c**. Cell bodies are fitted with an ellipsoidal shape shown as a dashed line in **e**. Solid lines illustrate the construction used to measure the lever arm $\ell$. The scale bar is 5 μm for both **d,e**.

the time trace of the instantaneous speed of a single rotor obtained from the angle difference between successive frames. The speed displays rapid fluctuations with a s.d. amounting to 19% of the mean value. Fourier analysis (Fig. 4c) reveals a characteristic noise frequency of about 15 Hz which is compatible with the typical wobbling frequency of freely swimming bacteria[21]. If we apply a low pass filter with a cutoff at 10 Hz the remaining fluctuations are quite small (colour curves in Fig. 4b) and only give rise to a 4.9% broadening of the rotational speed in individual rotors.

In our design we use an axis to constrain lateral motions of the rotor and minimize hydrodynamic coupling with the solid substrate. However, the presence of the axis itself might introduce an extra friction term due to the sheared fluid in the gap between the rotor and the axis. To quantify the magnitude of these effects we measure the rotational drag of the rotors by tracking their angular Brownian motion in the absence of bacteria. The measured value of $9.4 \pm 0.2$ pN µm s rad$^{-1}$ is close to the theoretical value[22] of 8.6 pN µm s rad$^{-1}$ calculated for an oblate ellipsoid with equivalent aspect ratio and volume. This confirms that the rotor has a minimal rotational drag for its size, meaning that it is not affected significantly by the proximity of the coverglass surface or the axis. From the measured rotational drag we can evaluate that an external torque of 17.5 pN µm would be required to spin an empty rotor at the same speed as that observed in the bacterial bath. Since a rotor holds 13.5 bacteria on average, we find that a single cell provides an effective torque of 1.3 pN µm. By analysing the spatial arrangement of bacteria, as observed in fluorescence images like 2e), we extract the lever arm $\ell$ as the distance of the rotor centre from the line passing through the cells major axis. We find an average value of $\ell = 6$ µm resulting in an effective pushing force per cell that is 0.2 pN. This value is about half of the typical flagellar thrust reported in the literature for *E. coli* cells[23,24]. Although flagellar thrust can vary with strain, medium composition and other experimental conditions, a possible explanation for the low value obtained here could be that flagellar bundles extending out of the structures will generate a counter rotating flow that could result in an effective rotational drag that is higher than what measured during free Brownian motion. Moreover, when the rotor is loaded, small asymmetries in bacteria configuration also result in an applied net force that maintains the rotor in close contact with the axis. This could also result in an increased rotational drag.

When comparing to previous 2D bacterial ratchet motors[11,12] there are few quantitative considerations that are worth mentioning. We can easily achieve rotational speeds of 20 r.p.m. corresponding to a linear speed of the outer rotor edge of 16 µm s$^{-1}$, which is very close to the speed of freely swimming bacteria ($\sim 20$ µm s$^{-1}$) in our experiment. For 2D ratchet motors typical rotational speeds were 1 r.p.m. with a corresponding edge speed of 2.5 µm s$^{-1}$. 2D motors display large speed fluctuations (100%) both in time and among rotors. Here we observe that practically every motor rotates with an average speed that only fluctuates by 8% (s.d.) among rotors, and with time fluctuations of only 5% (when noise components faster than 10 Hz are filtered out). Finally 2D rotors required high bacterial concentrations ($10^{10}$ cell per ml) limiting the durability of these micromachines. In contrast 3D motors spin much faster in a much more diluted bacterial suspension ($10^8$ cell per ml).

**Tuning global rotational speed with light**. An important issue for bacteria driven micromotors is that the rotational speed is uncontrolled and can vary a lot depending on the motility characteristics of the actual bacterial bath. More importantly, bacterial motility drops down quite abruptly if the oxygen and the

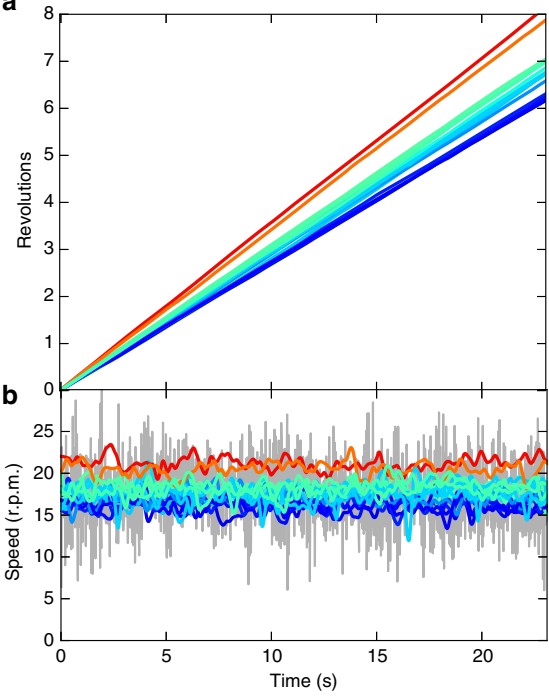
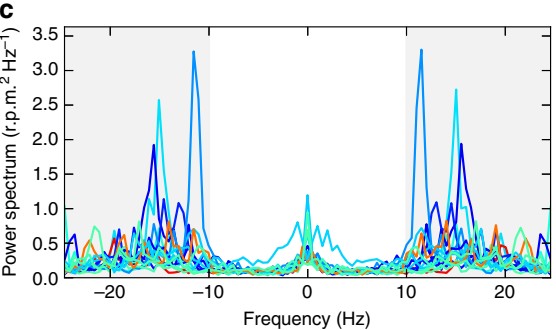
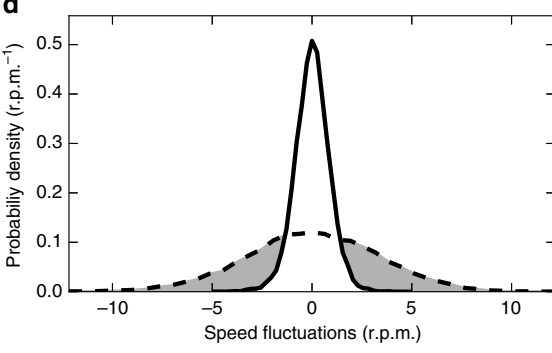

**Figure 4 | Characterization of rotational dynamics.** (**a**) Cumulative angle as a function of time for the 16 micromotors shown in Fig. 2b, colour scale encodes the average rotational speed from low (blue) to high (red). (**b**) Instantaneous rotational speed of the 16 rotors before filtering (grey line, only shown for one rotor) and after the low pass (10 Hz) frequency filtering (coloured lines, shown for all rotors). (**c**) Power spectra of the speed fluctuations. High frequency regions (>10 Hz) are marked with a grey background and filtered out. (**d**) Probability distribution of the fluctuations of the rotational speed before filtering (dashed line) and after filtering (full line).

energy-yielding nutrients are depleted from the medium by the cells[25]. This is a consequence of the fact that flagellar motors are driven by a flux of protons originating from the proton-motive force (PMF), an electrochemical gradient built up by the cell across its inner membrane through respiration. This limits the lifetime of bacteria powered micromotors in sealed samples that do not guarantee the constant supply of oxygen and nutrients required for cellular respiration. We demonstrate that this constraint can be overcome by exploiting proteorhodopsin (PR), a light-driven proton pump that uses photon energy to pump protons against the electrochemical gradient[26,27] across the cell membrane. By using PR expressing *E. coli* bacteria and a green light source, we are able to preserve cell motility even in a hermetically sealed sample. Furthermore the PMF, and consequently the swimming speed of the cells, can be thoroughly controlled by light intensity[28] enabling to externally tune the speed of micromotors.

We obtain light controllable micromotors by preparing sealed samples of microstructures in a suspension of PR expressing bacteria. We first characterize the rotational speed of self-assembled micromotors as a function of green light power. Figure 5a shows the speeds of a set of micromotors as the light intensity was systematically lowered in discrete steps while allowing enough time for the motors to reach a stationary speed at each light level. The datapoints are well fitted by a hyperbola, in accordance to the model of proteorhodopsin-generated PMF proposed in[27]. To characterize the dynamical response to light we then modulate the illumination intensity between 1.2 and 71 mW mm$^{-2}$ with a square wave having a period of 8 s. The resulting speed of a micromotor is plotted as a function of time in Fig. 5b and clearly shows the presence of a dynamical modulation. To better quantify the response raise and fall times we compute the average response over 10 periods (Fig. 5c). The average response is very well fitted by two exponentials having a rise time of 0.49 s which is significantly shorter than the fall time 0.73 s as could be expected from previous studies on single cell rotational dynamics[27].

**Energy considerations**. The mechanical power generated by each motor can be estimated by multiplying the measured rotational drag (9 pN μms rad$^{-1}$) by the square of the typical angular speed 1.9 rad s$^{-1}$ (18 r.p.m.). The value we get for the output mechanical power is ~30 aW per rotor. It is now interesting to compare this value with the rate of energy consumption. In the previously described experiments input energy is of two different kinds: chemical or optical. For chemically driven rotors, if we assume that glucose in the buffer is the primary energy source, we can estimate the rate of energy consumption by multiplying the glucose consumption rate per cell (~7 amol min$^{-1}$ per cell[25]) by the energy yield of glucose oxidation (3 × 10$^3$ kJ mol$^{-1}$) and by 15, the number of cells per rotor. The resulting consumption rate of chemical energy is about 5 pW. The conversion efficiency from chemical energy to mechanical work is therefore of order 10$^{-5}$. In light-driven rotors, when both glucose and oxygen are absent, the primary source of energy comes from photons which impinge on the structure with a typical energy flux of a few mW mm$^{-2}$ (Fig. 5a). The total optical power flowing through a single structure of radius 8 μm is then of order μW resulting in an efficiency of light-to-work conversion of order 10$^{-11}$. Although this is a very low efficiency, still it is much larger than what obtained when rotating microscopic objects through direct transfer of optical momentum[29].

**Closed loop control of individual micromotor speeds**. We have shown that light can be used to switch motors on and off and to tune their rotational speed. Using light as the control field has the enormous advantage that light can be shaped in time and space

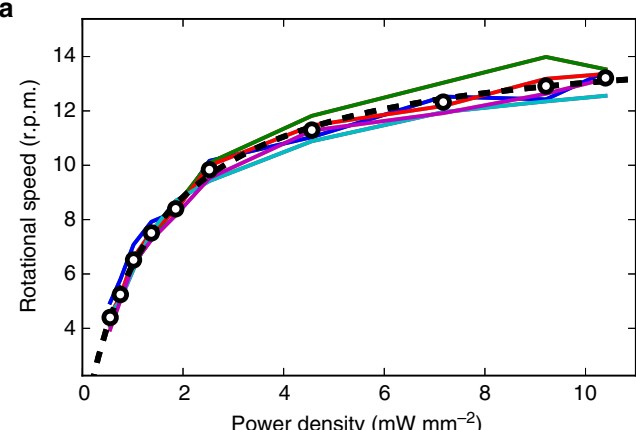

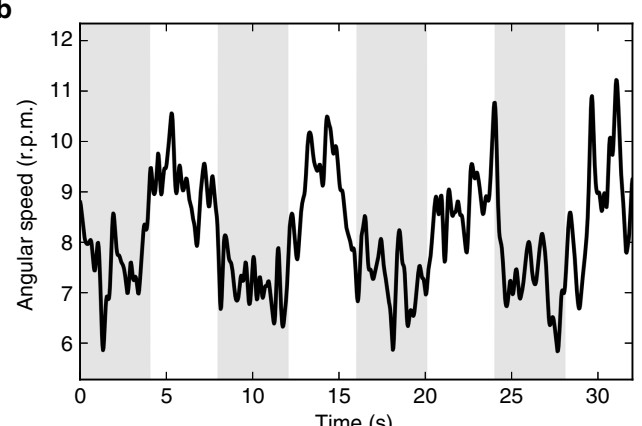

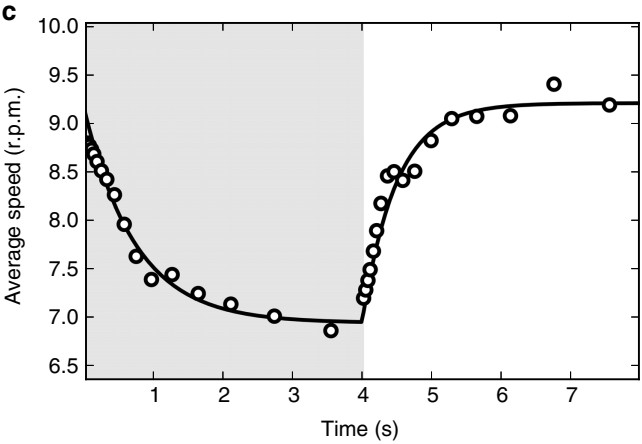

**Figure 5 | Light modulation of rotational speed.** (**a**) Solid lines represent the rotational speed of 5 micromotors obtained by progressively lowering the illumination power. The power dependence of the 5 rotor average speed (open circles) is very well fitted by a hyperbola (dashed line). (**b**) Dynamic response of rotational speed (full line) to a square wave-modulated light intensity (8 s period). The half periods with low light are represented with a grey background. (**c**) Rotational speed averaged over 10 periods (points), the full line represent a fit with two exponentials.

with a much higher precision than what is attainable with other possible control strategies such as by modulating the chemical environment[12]. This allows to rapidly and independently address individual rotors by projecting on each structure an illumination disk with variable intensity. We produce such structured illumination patterns by coupling a Digital Light Processor

(DLP) to a custom built optical microscope. We start by illuminating six target structures with six illumination disks having the same light intensity. We then track the angular motion of the motors and consequently update the 6 light levels with a real-time feedback loop. The feedback algorithm is such that the illumination level of the $i$-th disk at iteration step $n+1$ is:

$$I_{n+1}^i = I_n^i v_0 / v_n^i \qquad (1)$$

where $v_0$ is a target rotation frequency while $v_n^i$ is the frequency of the $i$-th rotor at iteration step $n$. The algorithm clearly converges when all rotors spin with the same frequency $v_0$. Before the loop switches on the rotors are uniformly illuminated and their rotational speeds range from 6.5 to 9.0 r.p.m.s. The feedback loop starts at $t=0$ and executes one iteration every 10 s (vertical dashed lines in Fig. 6). After only three iterations, the s.d. of the micromotor speeds is reduced from 1 r.p.m. to 0.2 r.p.m. that is from 13 to 2.7%, as can be seen in Fig. 6b). This small dispersion is maintained as long as the feedback loop is running and requires constant adjustments of the light levels (Fig. 6c).

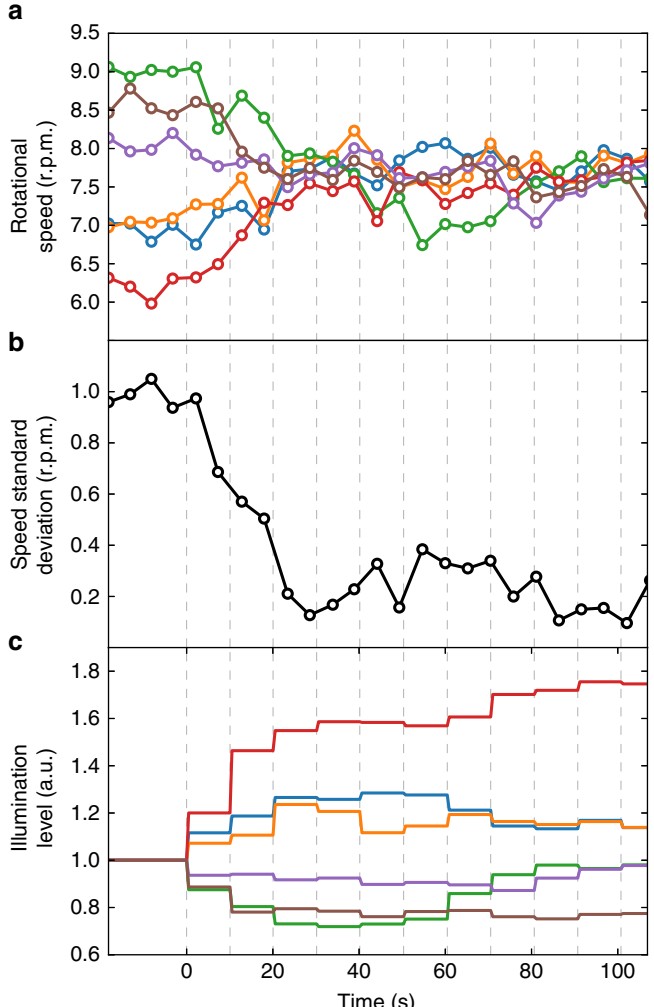

**Figure 6 | Closed loop control of individual micromotor speeds.**
(**a**) Rotational speeds of 6 micromotors driven by light powered bacteria. A feedback control loop is turned on at $t=0$ and adjusts light levels on each rotor based on its current speed. The feedback loop operates with a 10 s time interval shown as vertical dashed lines. (**b**) The s.d. of the speed in the 6 rotor sample quickly drops when we switch on the feedback control ($t=0$). (**c**) Solid lines represent the light levels over each micromotor (colour coding same as in **a**).

## Discussion

We show that fast, low-noise and light-controlled bio-synthetic micromotors can be obtained via the self-assembly of 3D microstructures and genetically engineered swimming bacteria. Using multifocal direct laser writing we produce large arrays of composite microstructures having submicron features that guide and arrange individual cells into ordered configurations. Trapped cells exert a stable and highly reproducible torque on the rotors. When propelling bacteria express the light-driven proton pump proteorhodopsin, the motors can be embedded in sealed environments where light provides the required energy supply. Using light patterns that are shaped in space and time with a DLP, we can control the speed of individual micromotors by addressing them with distinct light levels. We design a feedback control loop that automatically updates light levels based on the actual speeds of the rotors obtained by real-time video tracking. This allows to drive synchronously multiple micromotors and compensate for speed fluctuations in real-time. Future studies may try to explore the role of hydrodynamic coupling between nearby rotating units. This could lead to higher speeds when using different rotor lattices with alternating rotational directions. Differently from previous work on synchronization of rotors[30], our micromotors are torque free. Consequently hydrodynamic interactions will have a shorter range but also a different structure. Investigating hydrodynamic couplings in these self-propelled systems could reveal novel synchronization effects that could be systematically studied by exploiting the independent light tunability of coupled rotors.

## Methods

**Microfabrication.** Microfabrication was carried out by a custom built two-photon polymerization setup (Supplementary Fig. 1). The setup is based on a near infrared femtosecond fibre laser (FemtoFiber pro NIR, TOPTICA Photonics AG) with 780 nm wavelength, 87 fs pulse duration, 80 MHz repetition rate and 160 mW optical power. Exposure during fabrication is toggled by an optical shutter (SH). The fabrication laser power is set by a rotatable half-wave plate followed by a polarizing beam splitter cube (PBS). After expansion by lenses $L_1$ and $L_2$ the laser beam is reflected onto a holographic spatial light modulator (SLM) (X10468-02, Hamamatsu Photonics), which is in 4f conjugation to the back focal plane of a high numerical aperture oil immersion objective (Nikon Plan Apo Lambda 60x 1.4) by lenses $L_3$ and $L_4$. The SLM is used to generate multiple fabrication foci and to impose wavefront correction on the fabrication beam. The zero and the high diffraction orders can be blocked in the focal plane of $L_3$ by a thin wire and by an adjustable rectangular aperture (A). During fabrication the high NA focus of the laser is scanned inside a photoresist layer (S) carried on a microscope coverglass. Scanning is done by a 3-axis piezo translation stage (P563.3CD, Physik Instrumente (PI) GmbH & Co. KG) controlled through a NI-DAQ DA card. The micromotor structures were created from SU-8 2015 photoresist (MicroChem Corp). Four focal spots arranged in a 40 μm square were created with the SLM to increase the fabrication throughput. After exposure the photoresist sample was baked at 100 °C for 7 min, then developed by its standard developer solvent and finally rinsed in a 1:1 mixture of water and ethanol. Strong adhesion of the structures to the carrier coverglass was ensured by a layer of OmniCoat adhesion promoter (MicroChem Corp). Laser power and scanning speed varied between 1.9 to 5 mW per focus and between 24 and 100 μm s$^{-1}$, respectively.

**Microscopy.** Bright filed and epifluorescence imaging were performed on an inverted optical microscope (Nikon TE-2000U) equipped with a 60 × (NA = 1.27) water immersion objective and a high-sensitivity CMOS camera (Hamamatsu Orca Flash 4.0). For the periodic speed modulation of micromotors propelled by light-harvesting E. coli we powered the cells with the same high-power LED (Thorlabs M565L3) used for epifluorescence illumination. Independent control of micromotors was achieved by projecting the chip of a Digital Light Processor (Texas Instruments DLP Lightcrafter 4500) with the microscope imaging objective.

**Cell growth and sample preparation.** For these experiments we used the smooth swimming E. coli strain HCB437 (ref. 31). RFP + cells express the red fluorescent protein mRFP1 under the control of the lacI promoter (BioBricks, BBa_J04450 coding device inserted in pSB1C3 plasmid backbone, http://parts.igem.org/Catalogue). PR + cells express the light-driven proton pump, SAR86 γ-proteorhodopsin under the control of the araC-pBAD promoter (BioBricks, BBa_K1604010 coding device inserted in pSB1C3 plasmid backbone). Single colonies of RFP + and PR + cells were inoculated in 10 ml of LB medium and tryptone broth (TB) respectively, before growing overnight at 33 °C. The saturated culture was then diluted 1:100 (50 μl in

5 ml) into TB fresh medium and grown up to $OD_{590} \approx 0.8$ at 33 °C shaken (for aeration) at 200 r.p.m. The production of mRFP1 was induced during the last growth stage by addition of 1 mM IPTG. For PR + cells, 5 mM arabinose and 20 μM ethanolic *all-trans*-retinal were added to ensure expression and correct folding of PR in the membrane. In all culturing stages 25 μg ml$^{-1}$ kanamicyn and 34 μg ml$^{-1}$ chloramphenicol were present. Bacterial cells were then collected from culture media by centrifugation at 1,100$g$ for 5 min at room temperature. The pellet was resuspended by gently mixing in motility buffer (10 mM potassium phosphate (pH 7.0), 0.1 mM EDTA (pH 7.0), and 0.2% Tween 20)[32]. To increase the speed of the RFP + cells, glucose 10 mM was added to the motility buffer. The cells were washed three times to replace growth medium with motility buffer. Motility buffer sustains bacterial motility but not growth/replication so that the bacterial population remains constant during the experiment.

For experiments with PR + bacteria hermetically sealed samples were obtained using vacuum grease and thin nylon filaments as spacers. In these chambers oxygen is depleted by bacteria in less than half an hour at the cell concentration used in our experiments. After that, cells are non motile and only swim in the presence of green light illumination.

**Data availability.** The data that support the findings of this study are available from the corresponding author on reasonable request.

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

## Acknowledgements

We acknowledge S. Mansy for providing us the BBa_K1604010 plasmid. The research leading to these results has received funding from the European Research Council under the European Union's Seventh Framework Programme (FP7/ 2007–2013)/ERC Grant Agreement No. 307940.

## Author contributions

G.V., G.F. and R.D.L. designed experiments. G.V. and G.F. performed experiments. G.V. designed and fabricated microstructures. G.F. and F.S. were responsible for the transformation and growth of bacterial strains. C.M., G.V. and S.B. analysed the data. G.V., S.B. and R.D.L. designed and built the optical setups. G.V., C.M. and R.D.L. wrote the manuscript.

## Additional information

**Competing interests:** The authors declare no competing financial interests.

**Publisher's note**: 

