## [Peer Review File · Nature Communications]

Reviewers' comments:

Reviewer #1 (Remarks to the Author):

In this manuscript, the authors present a proof-of-principle implementation of rotors that are driven by bacteria with stable angular speeds. In addition, the authors show that by engineering the bacteria such that their motility can be controlled via light (using the proteorhodopsin approach), they have an accurate control over the angular speed of the rotors and can even implement a feedback loop such that many neighboring rotors can be rotated at the same speed.

This short manuscript is easy to read, the figures are clear and well-explained, and the overall findings are of broad interest due to their interdisciplinary scope, combining physics, engineering, and biology. So I think that Nature Communications is an appropriate journal for this manuscript. However, the manuscript currently has a big gap (problem 1, below), which I think should be closed before publication.

Problem 1: The authors do not really describe how they designed the 3D micromotors. Only one paragraph in the manuscript focuses on the design, giving only a few core dimensions of the motor. The manufacturing of the motors using 2-photon polymerization is well-described, but how the authors arrived at the particular design that they then use/characterize in this manuscript is unclear. Questions that come to my mind are:

- What is the angle of the ramp and why?
- What is the angle of the ridges in the ramp and why?
- Why does the rotor have 4 cutouts between the little chambers for the bacteria and the central hole for the shaft?
- What is the angle of the small chambers for the bacteria with respect to the tangent (or radius) of the rotor and why?

Problem 2: The authors mention (page 2, column 2, first paragraph) that "the collection efficiency is very high, about 90% of the microchambers are occupied [by bacteria]". It is unclear for which cell density in the medium the authors obtain such a collection efficiency. And: is the occupancy of the microchambers directly related to the bacterial cell density in the medium? Does "collection efficiency" refer to the fraction of bacteria that run up the ramp and are then captured by the microchambers? Or is it simply the fraction of occupied microchambers in the rotor? After which time is the occupancy of the microchambers measured? Is occupancy a quantity that reaches a steady state?

Suggestion (3): Can the authors make an estimate of the efficiency of such a motor? From chemical energy to useful mechanical work?

In summary, the authors currently present a proof-of-principle design of a bacteria-powered micromotor without giving a substantial description of the design principles of the motor. Instead, the authors focus (Figs. 4, 5) on tuning the rotation speed of the motor using the light-mediated control, which is also interesting, but only half the story without the design principles of the motor.

Once the authors address points 1, 2, and perhaps point 3 above, which will significantly strengthen the paper, I recommend publication.

Reviewer #2 (Remarks to the Author):

The work submitted by Di Leonardo and co-workers reports on microengineered motors powered by a bacterial bath and controlled by light beams. They use bacterial swimmers which are integrated into artificial and specially designed rotors. The work is very impressive from the

engineering point of view as the authors carefully and elegantly designed structures (micromotors) and coupled with illumination with tunes the speed of motors in a close loop manner. Actually, the most impressive part of the work is the close-loop control of individual motors which reduces the standard deviation with time. One have to say, otherwise, that the coupling of self-propelled swimmers (bacteria in this case) with microstructures is not the main novelty here as the group and others have previously reported it before. Yet, that is not a reason at all for not publishing. Indeed, it is positive as it continues the hybrid motors field and goes beyond the state of the art in a topic which I considered of high importance and general interest in multidisciplinary fields. I strongly recommend this paper for publication in Nat.Comm.

Some comments on points which could be more elaborated or discussed:

1. Is there any hydrodynamic coupling that exists between the rotors (micromotors) that leads to synchronization effects?
2. If not, could it be induced by using their feedback loop system?
3. IS it possible to look at the flow field around the rotors by using tracer particles for that ?
4. Could authors elaborate more the discussion on quantification (page 3, left column), at least in the Supp. Info?
5. Page 4. Authors mention that quantification is performed on 2 micromotors. However from figure 2 is clear that they can produce many more. Why are only 2 selected? Are they representative? More data for more statistics would be important.

Some minor comments:

1. Page 2. Figure 1d is cited before than a, -c. That is odd.
2. Page 2. Right column: "The speed displays rapid fluctuations on top of a rather flat baseline" sounds strange.
3. Can authors have a "c, not the free-swimming bacteria as seen in videos

Reviewer #3 (Remarks to the Author):

This manuscript presents an array of light controlled micro-motors powered by swimming E.coli. It integrates a micro-fabricated micro-motor with a 3D architecture powered by engineered bacteria. The presented micro-motor runs smoothly up to 16 rpm, its speed can be controlled by light with a feedback loop. The contribution of this work includes the tight control of motor rotation speed and no requirement of an water/air interface for the micro-motors to run on. A minor concern is the word '3D' in its title which can potentially be misleading. It is true that the architecture of the micro-motor is 3D, however, all the motors in the array are confined a few micrometers above a substrate, and are not free to move in three dimensional space at its current setting. There is a potential for this motor to work in 3D, however, it is not a true 3D motor in its current state. In summary, to gain a tight control of motor speed is a significant advancement in the development of bacterial powered micro-motor, and will be of great interest to biological as well as engineering community, and I recommend the manuscript to be published in Nature Communication. A few other minor concerns.

(1) A comparison of the rotational speed fluctuation of the current motor and the previous micro-ratchets developed in the same group will help the audience to appreciate the advancement of the presented work.

(2) Provide a scale bar in Figure 1.

(3) Provide captions for the supplementary movies.

Mingming Wu
Cornell University

Response to Reviewer #1:

In this manuscript, the authors present a proof-of-principle implementation of rotors that are driven by bacteria with stable angular speeds. In addition, the authors show that by engineering the bacteria such that their motility can be controlled via light (using the proteorhodopsin approach), they have an accurate control over the angular speed of the rotors and can even implement a feedback loop such that many neighboring rotors can be rotated at the same speed.

This short manuscript is easy to read, the figures are clear and well-explained, and the overall findings are of broad interest due to their interdisciplinary scope, combining physics, engineering, and biology. So I think that Nature Communications is an appropriate journal for this manuscript.

We thank the Reviewer for appreciating our work and supporting it for publication in Nat. Comm. Here follows a detailed list of replies to his/her comments

However, the manuscript currently has a big gap (problem 1, below), which I think should be closed before publication.

Problem 1: The authors do not really describe how they designed the 3D micromotors. Only one paragraph in the manuscript focuses on the design, giving only a few core dimensions of the motor. The manufacturing of the motors using 2-photon polymerization

is well-described, but how the authors arrived at the particular design that they then use/characterize in this manuscript is unclear. Questions that come to my mind are:

- What is the angle of the ramp and why?*
- What is the angle of the ridges in the ramp and why?*
- Why does the rotor have 4 cutouts between the little chambers for the bacteria and the central hole for the shaft?*
- What is the angle of the small chambers for the bacteria with respect to the tangent (or radius) of the rotor and why?*

Following the Reviewer's suggestion we now describe and discuss in greater detail the geometrical characteristics of the micromotors also including all of the important missing information pointed out by the Reviewer. For clarity we also added a new panel b) in Fig.1 showing a top view that clearly illustrates the principle of the design. Section **3D micromotor design** in **Results and discussion** has been now largely rewritten and expanded as follows:

Figure 1a shows the computer model of our micromotor design. The structure has three component parts. The rotating unit (appearing in green) has an external radius of 7.6 μm and a thickness of 3.7 μm . Its outer rim features 15 microchambers, each capable of accommodating one single cell body while leaving the entire flagellar bundle outside for maximal propulsion. The number of chambers a single rotor could carry is limited by the constraint that each chamber should be wide enough to fit one cell. Therefore the number of chambers is maximal when they are radially oriented but this configuration would also result in a zero torque. If we tilt the chambers by an angle θ the torque exerted by each cell will increase with $\sin\theta$ while the number of chambers will decrease as $\cos\theta$ (see Supplementary information). As a result the total torque would go as $\sin\theta \cos\theta$ and have a maximum at $\theta = 45^\circ$ which is the actual angle chosen in our design. In order to reduce fabrication time, the outer ring of the rotor is connected by four radial spokes to a central ring that is free to rotate around a vertical axis shown in blue. The top part of the axis has a wider cross-section to prevent rotor escape. A bottom platform on the axis keeps the rotor at a minimal height of $\sim 3 \mu\text{m}$, thus reducing hydrodynamic friction with the solid substrate. Moreover, the elevated position keeps the rotor out of the high bacterial concentration layer that forms on the substrate surface [15–17]. There, random collisions with free swimming bacteria would disturb the rotor's operation making it slower and less stable. To facilitate the capture of bacteria by the rotor, we built a radial ramp structure (red component) that collects bacteria swimming on the bottom surface and directs their trajectories upwards to the rotor's microchambers. However, with a bare ramp only a fraction of bacteria would be sent along trajectories that reach the tilted microchambers with the correct orientation to ingress easily. To overcome this problem we placed a series of barriers on the ramp. Incoming bacteria will align and slide along the barrier leaving the ramp on a trajectory that is very likely to intercept the rotors edge with

an angle matching the orientation of the microchambers (see dashed white line on Fig. 1a,b). All experiments were performed with non tumbling strains (smooth swimmers) which can be stably trapped in an empty hole by a constant flagellar thrust. We used two-photon polymerization [18,19] to fabricate micromotors from SU-8 photoresist on top of a microscope coverglass. Details of the actual structures, as seen by scanning electron microscopy (SEM), are shown in Fig. 1a) and 1b).

Problem 2: The authors mention (page 2, column 2, first paragraph) that “the collection efficiency is very high, about 90% of the microchambers are occupied [by bacteria]”. It is unclear for which cell density in the medium the authors obtain such a collection efficiency.

And: is the occupancy of the microchambers directly related to the bacterial cell density in the medium?

Does “collection efficiency” refer to the fraction of bacteria that run up the ramp and are then captured by the microchambers? Or is it simply the fraction of occupied microchambers in the rotor?

After which time is the occupancy of the microchambers measured?

Is occupancy a quantity that reaches a steady state?

The Reviewer is raising a number of very interesting points here that motivated us to perform new experiments directly aiming at elucidating the dynamics of microchambers loading. New data are now reported in Fig. 3 of the revised manuscript and discussed in the main text as follows (section **Self assembly of hybrid micromotors**):

We first immerse the structures in clean motility buffer (~ 500 μ L) and make sure that none of the rotors are stuck to the supporting axis. We then add ~ 100 μ L of a dense (OD₅₉₀=0.8) cell suspension resulting in typical surface densities of cells over the coverslip of about 0.015 cells/ μ m². Within few minutes after the addition of bacteria, the rotors start to capture cells and rotate. This self-assembly mechanism is very robust, practically every single rotor spins even in large and dense ensembles such as the array of 36 rotors shown in Fig. 2a and Suppl. Movie 3. By transforming a RFP expressing plasmid in our strain we can use epifluorescence microscopy to visualize the cell bodies captured inside the microchambers (Fig. 2c,e and [Suppl.Movie 1-2]). To characterize the dynamics of this self assembly process we recorded a sequence of bright-field clips (2 s) intercalated by short clips (0.1 s) in epifluorescence mode throughout the entire fill up process. Bright field clips are used to track rotational speeds of individual rotors while simultaneously keeping track of the corresponding number of trapped cells observed in fluorescence images. Fig. 3a shows the number of captured bacteria as a function of time. The average number of occupied microchambers (shown as red crosses) reaches the 90% of its maximal value (13.5) after only 2 minutes. The

overall time behavior can be very well fitted by an exponential law with a time constant of 49 s and a saturation value of 14.4 (96%). Furthermore, we find that rotational speeds increase linearly with the number of captured cells (Fig. 3b).

Suggestion (3): Can the authors make an estimate of the efficiency of such a motor? From chemical energy to useful mechanical work?

Following the Reviewer's suggestion we added a new section on energy considerations which is reported below:

The mechanical power generated by each motor can be estimated by multiplying the measured rotational drag ($9 \text{ pN } \mu\text{m s rad}^{-1}$) by the square of the typical angular speed 1.9 rad/s (18 rpm). The value we get for the output mechanical power is $\sim 30 \text{ aW}$ per rotor. It is now interesting to compare this value with the rate of energy consumption. In the previously described experiments input energy is of two different kinds: chemical or optical. For chemically driven rotors, if we assume that glucose in the buffer is the primary energy source, we can estimate the rate of energy consumption by multiplying the glucose consumption rate per cell ($\sim 7 \text{ amol/min/cell}$ ²⁵) by the energy yield of glucose oxidation ($3 \cdot 10^3 \text{ kJ/mol}$) and by 15, the number of cells per rotor. The resulting consumption rate of chemical energy is about 5 pW . The conversion efficiency from chemical energy to mechanical work is therefore of order 10^{-5} . In light-driven rotors, when both glucose and oxygen are absent, the primary source of energy comes from photons which impinge on the structure with a typical energy flux of a few mW/mm^2 (Fig. 5a). The total optical power flowing through a single structure of radius $8 \mu\text{m}$ is then of order μW resulting in an efficiency of light- to-work conversion of order 10^{-11} . Although this is a very low efficiency, still it is much larger than what obtained when rotating microscopic objects through direct transfer of optical momentum²⁹.

In summary, the authors currently present a proof-of-principle design of a bacteria-powered micromotor without giving a substantial description of the design principles of the motor. Instead, the authors focus (Figs. 4, 5) on tuning the rotation speed of the motor using the light-mediated control, which is also interesting, but only half the story without the design principles of the motor. Once the authors address points 1, 2, and perhaps point 3 above, which will significantly strengthen the paper, I recommend publication.

Following the Reviewer's suggestions we have expanded the manuscript with a deeper and more detailed description of the design principles using text, new data and new figures. We have addressed all 3 points raised by the Reviewer resulting in a revised manuscript that is much stronger in terms of clarity and experimental evidence.

Response to Reviewer #2:

The work submitted by Di Leonardo and co-workers reports on microengineered motors powered by a bacterial bath and controlled by light beams. They use bacterial swimmers which are integrated into artificial and specially designed rotors. The work is very impressive from the engineering point of view as the authors carefully and elegantly designed structures (micromotors) and coupled with illumination with tunes the speed of motors in a close loop manner. Actually, the most impressive part of the work is the close-loop control of individual motors which reduces the standard deviation with time. One have to say, otherwise, that the coupling of self-propelled swimmers (bacteria in this case) with microstructures is not the main novelty here as the group and others have previously reported it before. Yet, that is not a reason at all for not publishing. Indeed, it is positive as it continues the hybrid motors field and goes beyond the state of the art in a topic which I considered of high importance and general interest in multidisciplinary fields. I strongly recommend this paper for publication in Nat.Comm.

We thank the Reviewer for recommending our work for publication in Nat. Comm. Here follows a detailed list of replies to his/her comments

Some comments on points which could be more elaborated or discussed:

1. Is there any hydrodynamic coupling that exists between the rotors (micromotors) that leads to synchronization effects?
2. If not, could it be induced by using their feedback loop system?
3. IS it possible to look at the flow field around the rotors by using tracer particles for that ?

The issue of hydrodynamic coupling in self-propelled systems is actually very interesting and poorly explored in current literature that mainly focuses on externally driven systems. We now comment more on the nature of the hydrodynamic coupling in our systems and on related possible perspectives in the concluding section.

Future studies may try to explore the role of hydrodynamic coupling between nearby rotating units. This could lead to higher speeds when using different rotor lattices with alternating rotational directions. Differently from previous work on synchronization of rotors [30], our micro-motors are torque free. Consequently hydrodynamic interactions will have a shorter range but also a different structure. Investigating hydrodynamic couplings in these self propelled systems could reveal novel synchronization effects and could be systematically studied by exploiting the independent light tunability of coupled rotors.

We also thank the reviewer for offering two actually very interesting suggestions. Using tracer particles is definitely possible and we are already working in this direction. Achieving phase

locking with the feedback loop probably requires a dynamical response of bacteria to light that is faster than what we have now. We are currently investigating strategies to reduce this response time.

4. Could authors elaborate more the discussion on quantification (page 3, left column), at least in the Supp. Info?

Following suggestions from Reviewer 2 (and also 1) we have improved a lot the quantitative discussion of the results adding:

1. Details and working principles of the micro-motors design
2. A detailed characterization of the self-assembly dynamics performing new experiments summarized in the new Fig. 3
3. A detailed and quantitative comparison with previous 2D designs (see reply Reviewer 3 point 1)
4. A quantitative discussion on the energetic efficiency has been added in the new section

Energy considerations

For more details please see reply to Reviewer 1

5. Page 4. Authors mention that quantification is performed on 2 micromotors. However from figure 2 is clear that they can produce many more. Why are only 2 selected? Are they representative? More data for more statistics would be important.

Collecting full speed vs light curves can be problematic because when the light level, and thus flagellar thrust, is too low, bacteria often escape from the microchambers so that many curves are incomplete. However we decided to repeat the experiment to improve the statistics. Fig. 5 now reports data for 5 rotors which display a very consistent behavior throughout the light intensity range. We thank the Reviewer for giving us the chance to improve our results on that point.

Some minor comments:

1. Page 2. Figure 1d is cited before than a,-c. That is odd.

Fig. 1 is now changed including new and better SEM images and a second view on the 3D design. The new panels in Fig. 1 now appear in the same order as in the text.

2. Page 2. Right column: "The speed displays rapid fluctuations on top of a rather flat baseline" sounds strange.

We now say:

The speed displays rapid fluctuations with a standard deviation amounting to 19% of the mean value.

3. Can authors have a “c, not the free-swimming bacteria as seen in videos

There must be a typo in this comment, we unfortunately cannot understand this question. We hope this was just a minor remark.

Response to Reviewer #3:

This manuscript presents an array of light controlled micro-motors powered by swimming *E.coli*. It integrates a micro-fabricated micro-motor with a 3D architecture powered by engineered bacteria. The presented micro-motor runs smoothly up to 16 rpm, its speed can be controlled by light with a feedback loop. The contribution of this work includes the tight control of motor rotation speed and no requirement of an water/air interface for the micro-motors to run on. A minor concern is the word '3D' in its title which can potentially be misleading. It is true that the architecture of the micro-motor is 3D, however, all the motors in the array are confined a few micrometers above a substrate, and are not free to move in three dimensional space at its current setting. There is a potential for this motor to work in 3D, however, it is not a true 3D motor in its current state. In summary, to gain a tight control of motor speed is a significant advancement in the development of bacterial powered micro-motor, and will be of great interest to biological as well as engineering community, and I recommend the manuscript to be published in Nature Communication.

We thank the Reviewer for recommending our work for publication in Nat. Comm. Using a 3D architecture has allowed us to overcome many limitations that are unavoidable in 2D (precise control of number, position and orientation of bacteria, reduced friction, fixed axis and more...). For this reason we wanted to highlight the 3D character of our structure in the title. The revised manuscript now includes a more detailed description (see reply to Reviewer 1 point 1) of the working principle of the 3D design which we hope will help appreciating the importance of 3D fabrication. Moreover, following Reviewer 3 suggestion (see point 1 below), we now discuss in quantitative detail all the improvements obtained by moving from a 2D to a 3D architecture. However if the Reviewer still believes that the word 3D in the title is really misleading we can remove it.

A few other minor concerns.

(1) A comparison of the rotational speed fluctuation of the current motor and the previous micro-ratchets developed in the same group will help the audience to appreciate the advancement of the presented work.

We thank the Reviewer for this valuable suggestion. That part was actually missing and we now included a detailed quantitative comparison with our previous design:

When comparing to previous 2D bacterial ratchet motors^{11,12} there are few quantitative considerations that are worth mentioning. We can easily achieve rotational speeds of 20 rpm corresponding to a linear speed of the outer rotor edge

of 16 μm , which is very close to the speed of freely swimming bacteria ($\sim 20 \mu\text{m/s}$) in our experiment. For 2D ratchet motors typical rotational speeds were 1 rpm with a corresponding edge speed of $2.5 \mu\text{m/s}$. 2D motors display large speed fluctuations (100%) both in time and among rotors. Here we observe that practically every motor rotates with an average speed that only fluctuates by 8% (s.d.) among rotors, and with time fluctuations of only 5 % (when noise components faster than 10 Hz are filtered out). Finally 2D rotors required high bacterial concentrations (10^{10} cell/mL) limiting the durability of these micro-machines. In contrast our 3D motors spin much faster in a much more diluted bacterial suspension (10^8 cell/mL).

(2) Provide a scale bar in Figure 1.

We thank the Reviewer for pointing this out, there were actually other figures with missing scale bars, we have now corrected for that.

(3) Provide captions for the supplementary movies.

The following captions were inserted in the online submission system:

Supplementary video 1: Array of 16 microrotors actuated by smooth swimming bacteria. The movie switches from bright field mode to epifluorescence revealing the high occupancy number of microchambers.

Supplementary video 2: Close view of a fully loaded micromotor pushed by 15 bacteria as revealed in the second half of the video where we switch to epifluorescence.

Supplementary video 3: A dense array of 36 microrotors all spinning with a smooth and uniform speed.

REVIEWERS' COMMENTS:

Reviewer #1 (Remarks to the Author):

This is now an excellent manuscript, based on a creative idea and motor design, as well as a clear presentation and interesting figures. I recommend publication of the manuscript in its current form.

Reviewer #2 (Remarks to the Author):

The authors carefully replied to all the comments from the referees. To this point, I am very satisfied with this revision which has improved the quality of the paper and it should be ready for publication in its current form. I congratulate the authors for their work.

Reviewer #3 (Remarks to the Author):

The authors have adequately addressed my concern.